# Effects of Within-Storm Variability on Allochthonous Flash Flooding: A Synthetic Study

**Shahin Khosh Bin Ghomash** [1,2,*] **, Daniel Bachmann** [1] **, Daniel Caviedes-Voullième** [3,4,5] **and Christoph Hinz** [2]

1   Research Group Flood Risk Management, Magdeburg-Stendal University of Applied Sciences, 39114 Magdeburg, Germany
2   Chair of Hydrology, Brandenburg University of Technology Cottbus-Senftenberg, 03046 Cottbus, Germany
3   SimDataLab Terrestrial Systems, Jülich Supercomputing Centre, Forschungszentrum Jülich, 52428 Jülich, Germany
4   Institute for Bio-Geosciences, Agrosphere IBG-3, Forschungszentrum Jülich, 52428 Jülich, Germany
5   Centre for High Performance Scientific Computing Terrestrial Systems (HPSC-TerrSys), Geoverbund ABC/J, 52428 Julich, Germany
\*   Correspondence: shahin.khoshbinghomash@h2.de

**Abstract:** Rainfall is a spatiotemporally variated process and one of the key elements to accurately capture both catchment runoff response and floodplain extents. Flash floods are the result of intense rainfall, typically associated to highly variable rain in both space and time, such as convective storms. In this work, the extent within-storm variability affects runoff and flooding is explored. The Kan catchment (Tehran, Iran) is used as base topography for the simulations. The allochthonous nature of floods in the catchment and how they interact with the effects of storm variability are further investigated. For this, 300 synthetic rainfall signals with different hyetograph variabilities are generated and imposed on a 1D/2D hydrodynamic model. Additionally, a set of simulations with different levels of spatial variability are performed. The results suggest that temporal and spatial variability affect the runoff response in different degrees. Peak discharge and hydrograph shapes, as well as flooded areas, are affected. The effect of storm temporal variability is shown to be significantly higher than storm spatial variability and storm properties such as return period, duration, and volume. Further on the influence of storm spatiotemporal variability on stream discharge and flood response is seen to be strongly dependent on the location within the drainage network at which it is assessed.

**Keywords:** flash flood; rainfall variability; rainfall/runoff simulation; runoff generation

## 1. Introduction

Rainfall is a complex, spatial, and temporally variated process [1], and one of the core inputs for hydrologic and hydrodynamic flood modelling. Rainfall hyetograph shape and structure have a governing influence in modelling the flood response of river basins [2,3]. Moreover, increased extreme storms, with different spatial and temporal variability, are expected to occur in the future as a result of climate change [4,5]. Therefore, exploring the extent rainfall spatiotemporal variability may affect runoff and flooding is a valuable asset for adequate and effective river discharge estimation, flood extent estimation, and flood risk analysis.

Flash floods, a fast flooding of land adjacent to a river, can be caused by exceeding the local stream drainage capacity after severe rainfall events, and are modulated by several factors: terrain gradient, vegetation coverage, soil type, surface imperviousness, land cover, as well as precipitation [6,7]. Flash-flood-prone regions typically exhibit complex geometries which, in turn, induce complex and transcritical flow patterns [8] and can, thus, be demanding in terms of hydrodynamic modelling, used to investigate the dynamic and complex process of flash floods [8–10].

The hydrodynamic modelling of floods is one key component of flood risk management [11]. To achieve meaningful and realistic modeling results, rainfall features (e.g., peak intensity) which have a governing influence in the storm runoff response of catchments [12,13] must be adequately represented. This warrants precipitation data with high spatial and temporal resolutions [14]. In flash flood and urban flood risk assessment, design storms (correlated to a certain return period, and typically derived based on point rainfall records) have been often used and are often represented as spatially uniform precipitation [15]. This is mainly due to practical limitations such as low-density rain gauge networks, or merely the inherent simplicity of representing rainfall in a spatially uniform fashion in flood simulations. Though the use of such design storms is convenient, it can neglect the complex impact of within-storm temporal and spatial variability on the hydrological response of urban catchments [16,17]. Such variability has been identified as a major source of error in rainfall-runoff modelling [18]. Additionally, most of the available rainfall data usually has a temporal resolution which is arguably not sufficient for accurate flash flood and urban flood modelling [19]. This is evolving, with increasing availability of automated rainfall gauges and telematic data collection, providing higher temporal resolution [20], as well as remote sensing data.

An additional level of complexity is that in some catchments with relatively large areas, rainfall might take place only in the mountainous areas, while the flooding often manifests in the plains further downstream, in which little or no rainfall is observed during the flood event. This type of flooding may be characterized as *allochthonous* river flooding, due to the origin of the water being elsewhere. Previously some studies have focused on the allochthonous nature of rivers. Wrzesiński et al. [21] examined the flow regime alterations of the Vistual River (Poland) between the years 1971 to 2010 and found the allochthonous nature of the river to be a governing influence in these alterations. Deodhar et al. [22] linked the decrease in the width-depth ratio of the Deccan Trap rivers in India to changes in discharge due to the allochthonous nature of the rivers. Khosh Bin Ghomash et al. [23] studied the flash flood response of the Kan River (Iran) to storms moving with different velocities and directions and found that the allochthonous nature of the river highly influences runoff sensitivity to storm movement. While several flood events such as the flooding of the Elbe river in Magdeburg (Germany) in the year 2013 can be categorized as allochthonous [24], it is interesting to study to what extent within-storm temporal and spatial variability can influence and interact with such floods. This merits the question: how both the allochthonous nature of the flood and the runoff scale effects can interact with rainfall within-storm variability.

A number of studies have explored the influence of rainfall spatiotemporal variability on the hydrodynamic response of catchments. Faures et al. [25] found that spatial variability can translate into large variations of simulated runoff volume and peak discharge in small catchments (<5 ha). Zoccatelli et al. [26] reported that neglecting spatial variability can lead to a considerable loss of modelling efficiency in flash flood modelling, in a set of Romanian catchments (36–167 km$^2$). Lobligeois et al. [27] investigated the importance of detailed rain field mapping in France over a 10y period with a rainfall resolution of 1 km/1 h and emphasized the importance of rain spatial properties for hydrograph modeling. Gires et al., [28] observed differences of up to 40% in peak discharge caused by small scale (1 × 1 km, 5 min) rainfall variability in an urban catchment (900-ha) in London. Schellart et al. [29] found that rainfall spatial variability and data uncertainty can cause significant differences in simulated peak flows and their occurrence times in urban sewer flow estimates using rain gauge and radar data in an urban catchment (11 km$^2$) in England. Paschalis et al. [3] used stochastically generated rainfall ensembles as input for a hydrological model under different soil moisture conditions for a catchment in Switzerland. They found flood peak and the time of peak occurrence to be more sensitive to rainfall temporal variability rather than spatial variability and mostly influenced by the spatial clustering of saturated areas in the catchment. Several other studies have also focused on analyzing how runoff is affected by within-storm variability. The findings do not converge to a unified conclusion, and, in some cases, yield controversial results that emphasize the

complexity of the topic e.g., [30–35] and highlight that no clear consensus yet exists [17]. For example, Gabellani et al. [31] reported significant changes in the runoff response of the Tanaro basin in North-Western Italy due to rainfall spatiotemporal variability. In contrast, Smith et al. [30] suggest that rain spatial variability did not significantly influence the hydrological response of an urban catchment in Baltimore city. Clearly, these conclusions are case-dependent, and several factors have been identified to interact; for example, the relevance of variability can be modulated by antecedent soil saturation [36], infiltration capacity [37], and by averaging introduced by routing schemes [34]. A critical reading of the literature highlights the need for further detailed studies and more evidence of the possible effects of rainfall variability on flash flood response. In particular, the complexity of the interactions between spatial and temporal variability [16], how these effects are modulated by catchment properties [38], and the interactions of rainfall properties and the response of Hortonian runoff at different hydrological scales [39] remains a challenging topic.

Moreover, assessing the effects of rainfall variability through observed data can lead to apparently contradictory conclusions, which can be related to the difficulties in guaranteeing consistent rainfall volumes [18]. This may improve with increasing availability of dense rain gauge networks and radar data better representing spatial rainfall distributions. Nonetheless, this still poses the challenge of representing meaningful and consistent rainfall signals with variability, but which still allow for straightforward interpretation; in particular, to find, analyze, and understand links between the effects of variability and flood response. We focus in this work on the effects of within-storm variability on runoff generation and flood response. The Kan catchment (Teheran, Iran) is used as a case study of an allochthonous system. It is worthy to mention that the specific goal of this study is not representing nor reproducing a real flooding event, but instead conducting a theoretical comparative analysis of the influence of within-storm temporal and spatial variability on flooding via surface flow simulations. Although real flooding events and observations are crucial for a better understanding of such topics, synthetic modelling studies also play a key role. Observations are by their own nature constrained by the events, when and where they occur, making their replication and comparative analysis difficult and, in many cases, nearly impossible. Modelling is the only mean that enables a systematic sensitivity analysis of a system to variations in processes and to controlled forcing. Moreover, how the allochthonous nature of the flooding may affect the response, and how this influences the relationship between hydrographs observed at given points and flood response as a whole, are investigated. In contrast to most previous studies, in addition to the resulting hydrographs, we also focus on additional flow parameters, such as water depth and flood extent, which offer a more comprehensive assessment of the effect of within-storm variability, taking the conclusions a step closer to flood damage and risk analysis [40].

In order to facilitate the assessment of the effects of within-storm variability, some simplifying assumptions are made. Processes such as infiltration and roughness, which can greatly affect runoff and resulting hydrographs, also introduce complexity and uncertainty, thus lowering model interpretability [41]. Therefore, we construct a semi-idealized setup consisting of an impervious surface with homogeneous roughness to analyze model sensitivity to spatial and temporal within-storm variability, but nonetheless, we also compare results with simulations considering infiltration and roughness. The approach used in this study is mainly inspired by the downward approach [42,43] and has previously been used for studying the sensitivity of runoff generation to different processes in hydrodynamic modelling [23,44,45]. The main components of this study which build on previous works on the topic of rainfall spatiotemporal variability are as follows: (1). studying the effects of rainfall spatiotemporal variability on flash flooding and runoff in an allochthonous environment, which has not received much attention until now; (2). the construction of a controlled systematic modelling environment which enables studying the effects of rainfall spatiotemporal variability, independent of other processes, on surface flow and flooding (this setup also allows for rainfall volume consistency among the simulations which enables straightforward comparability); (3). in addition to the analysis of the resulting hydrographs in the simulations, also taking into

account other indicators such as flooded areas, which allow for a more comprehensive view and understanding on the effects of rainfall spatiotemporal variability.

The Kan catchment shows high potential for flash flood occurrence due to its steep topography, semi-arid climate and rocky surface, and has seen multiple flash flood incidents in the past decades [46,47]. Moreover, rainfall in the Kan catchment is mostly concentrated in the upper mountainous regions, north of Teheran, which trigger allochthonous flash floods in the urban floodplain [46,48,49]. Due to all of this, the Kan catchment offers an opportunity to study the influence of within-storm variability on the hydrodynamic modelling of flash floods.

The paper is structured as follows: in Section 2, descriptions of the catchment area, the hydrodynamic model, the model setup process, and detailed descriptions of the rainfall forcing are presented. Section 3 begins with the presentation and analysis of the effects of rainfall temporal variability and then puts focus on the effects of rainfall spatial variability. In Section 4, a review of the study and the main implications are highlighted.

## 2. Materials and Methods

### 2.1. The Catchment

The Kan catchment is located in the province of Tehran (Iran) with an approximate area of about 836 km$^2$. Figure 1 illustrates the boundaries and topography of the catchment. The catchment has a steep topography with an elevation ranging from 900 to 3800 m.a.s.l. It shows an overall average slope of 30 degrees from north to south. The main stream in the catchment is the Kan River, which has a length of approximately 72 km in the domain of interest. The catchment can be divided into three main areas. The upper catchment region (~200 km$^2$) is mainly characterized by high relief and steep mountainous terrain with narrow river valleys. This part of the catchment is generally covered by rock and brush ranch [49]. The middle catchment section, which corresponds to the metropolitan area of Tehran city, has an area of about 230 km$^2$ and has a relatively flat surface in comparison to the upper section the lower catchment area can be described as semi-urban, with patches of urban areas surrounded by agricultural fields. The topography of this section is comparable to the middle catchment area.

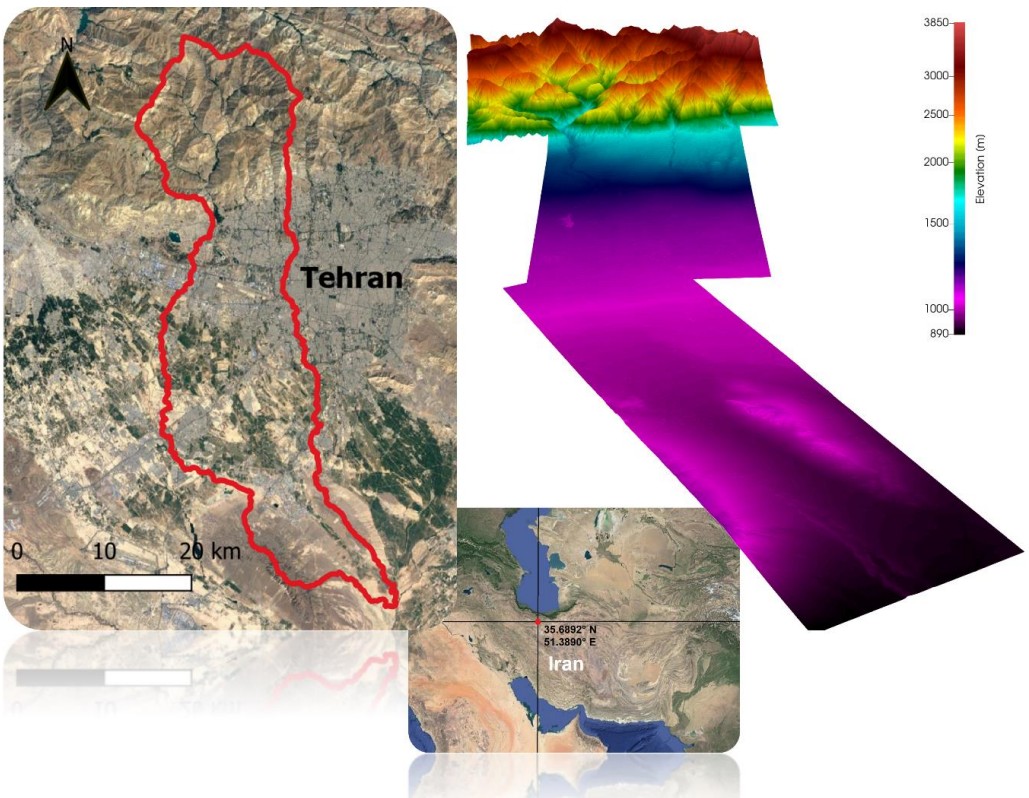

**Figure 1.** The Kan catchment. The red line shows the boundary of the catchment (**left**).

The area has a cold semi-arid climate [49,50]. This kind of climate tends to exhibit low precipitation, large variability in precipitation from year to year, and low relative humidity [51]. Rainfall spatiotemporal variability is known to be an important characteristic of semi-arid regions [52,53]. Semi-arid climates are also known to exhibit complex inter-annual patterns of rainfall spatial variability, but also during single rainfall events [53,54]. The climatic and topographic conditions of the Kan catchment favor a high potential for flash floods, with several occurrences reported in the last decades [46,47,55].

The annual rainfall and temperature in the catchment are strongly affected by the topographic gradient from north to south. According to Bokaie et al. [48], the annual precipitation in the catchment varies between a maximum of 422 mm in the north (upper catchment) and a minimum of 145 mm in the southeast (floodplains). In another report, the annual precipitation for the upper region of the catchment is reported to be 640 mm [56].

## 2.2. Hydraulic Model

Runoff dynamics, flood routing, and spreading were modeled using a hydrodynamic approach. The hydrodynamic module of the software package ProMaIDes—Protection Measures against Inundation Decision support—was applied. This software package is used for a risk-based assessment of flood protection measures for river floods, flash floods, and storm surges [57]. The hydrodynamic module of ProMaIDes was chosen due to its embedment in a broader flood risk analysis procedure, which enables the calculation of other flooding aspects such as damage and risk in future studies of the catchment area. The core hydrodynamic solver of ProMaIDes [58] solves the Zero-Inertia (ZI) approximation (Equation (1))—also often named as the diffusive wave approximation—to the shallow water equations [59,60]:

$$\frac{\partial h}{\partial t} + \nabla \left( \frac{h^{5/3}}{n\sqrt{\|Z\|}} Z \right) = R \tag{1}$$

where $h$ is water depth (m), $t$ is time (s), $n$ is Manning's roughness coefficient ($ms^{-1/3}$), $Z$ is the water surface gradient (-), $R$ in the 2D domain represents rainfall intensity (m/s) and in the 1D domain corresponds to the lateral inflow for each computation reach. The ZI equation has been extensively used and evaluated in flood simulation [61–64] and has been shown to be valid for rainfall-runoff simulation [65,66], used for Hortonian runoff in semi-arid regions [67] and flash flood modelling [68].

ProMaIDes implements the solution of the ZI equation in two distinct domains: 2D overland flow and 1D stream (channel) flow. Both 1D and 2D flow domains are solved by a first-order finite volume schemes with implicit time integration [58]. The 1D stream flow domains are discretized by cross-sections along the stream, and the 2D domains are discretized with regular grids. The 1D and 2D domains are then coupled explicitly in time, through mass flux exchange at defined coupling time intervals.1D-2D hybrid approaches have been shown to perform well for flood modelling e.g., [64,69–71]. For computational efficiency, ProMaIDes also allows to couple 2D-2D models (or subdomains), a feature which is leveraged here (Figure 2).

The simulations reported here were performed with ProMaIDes on an Intel Core i5-10400F CPU @ 2.90 GHz.

## 2.3. Study Setup

In the following, detailed descriptions of the steps taken in this study are presented. At first, in Section 2.3.1, the 1D-2D hydrodynamic model setup for the Kan catchment is described in detail. The model has been setup under two main setups: an idealized setup, designed to study the independent effects of rainfall spatiotemporal variability; and a non-idealized setup, implemented to assess the robustness of the results in the idealized scenarios. In Section 2.3.2, the boundary forcing for the hydrodynamic model are extensively explained. Three-hundred rain signals are generated using a rainfall model with different degrees of hyetograph temporal variability under different return periods and

durations to assess the effects of rain temporal variability. Further on, 50 of the generated rains are then given different degrees of spatial variability by means of a topographic-based approach in order to study the importance of rainfall spatial variability in the simulations.

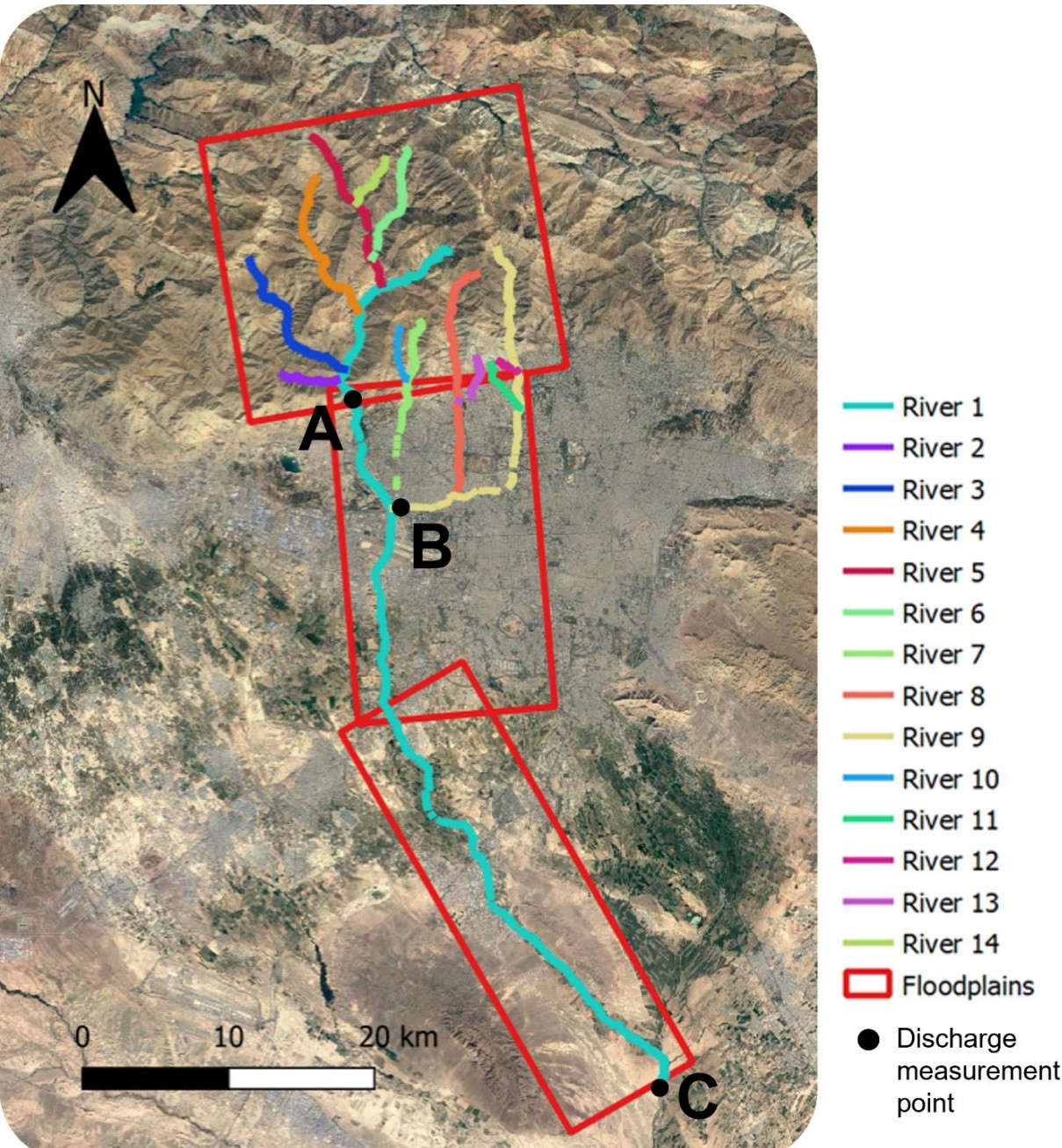

**Figure 2.** Overview of the 1d-2d hydrodynamic model for the Kan catchment. Three discharge measuring points during the simulations are shown. Point A (end of the upper subdomain) and C (end of the lower subdomain) are located on the Kan River itself, whereas point B is just upstream of the confluence of a tributary of the Kan River (i.e., does not gauge the discharge of the Kan River itself).

### 2.3.1. Model Setup

A spatially distributed 1D-2D hydrodynamic model was setup for this study. The model consists of a set of three subdomains (each represented in 2D by one regular grid) coupled with 14 1D-river models, illustrated in Figure 2. The topography in each subdomain

is based on the TanDEM-X 12 m digital elevation model. The 2D subdomains are discretized with a cartesian grid adding up to approximately 1.4 million cells. The upper subdomain (representing the mountainous regions of the catchment) has a resolution of dx = dy = 50 m and the two lower subdomains have a cell resolution of dx = dy = 25 m. Cell elevations were obtained by arithmetic averaging upscaling of the original 12 m DEM. Cells located outside the catchment boundary (Figure 1, red line) in the subdomains are set as non-computational cells during the simulations for further computational efficiency.

The 1D setup consists of 14 river models with a total of 1075 river cross-sections. The cross-sections were created with an average distance of 150 m. The cross-section elevation is also based on the TanDEM-X 12 m digital elevation model. The width of each cross-section is discretized by points with a 5 m resolution. The elevation in each point is computed by bi-linear interpolation of the elevation of the cell containing the point and its neighboring grid cells. With the help of satellite images and expert judgment, an intensive plausibility check of the cross-sections is performed. Although it is known that an adequate representation of river bathymetry is crucial in hydrodynamic modelling and may affect the available channel storage in the simulations [72], due to the data scarcity from the study area, a better representation of the river channels in this study was not possible and can be pointed out as a limitation. Nonetheless, the main focus of this study is not accurate flood extent estimation in the catchment, but observing the differences in runoff and flood response as a result of different storm spatiotemporal variabilities.

During the simulations, discharge is virtually measured at three different locations in the model. These measuring points are shown in Figure 2. Measuring point A is located in the main Kan river at the lower part of the upper sub-domain. Measuring point C is also situated in the main Kan River at the lowest part of the lower subdomain, illustrating how discharge is affected while going through both rain-induced and non-rain-induced domains. Measuring point B is placed at the end of River 9 (Figure 2), which is a main tributary of the Kan River. It is also worthy to mention that this measuring point is located at a position in which the contributing streams have more than half of their channels in the urban floodplain. Further on, the flood extents in each simulation are calculated by the number of flooded cells with water levels above a certain threshold, multiplied by cell area.

In the first step, inspired by the downward approach [42,43] and in order to facilitate the understanding of the effect of within-storm variability on runoff and flooded areas, we take the simplifying assumption of neglecting infiltration rates. To simplify the setup further on, a homogeneous roughness with a Manning's coefficient of 0.035 s/m$^{1/3}$ is introduced. Other processes such as subsurface flow and evaporation are neglected, since they occur at a different temporal scale. This is in line with the downward approach which implies to start with the simplest model structure and to increase the model complexity in a stepwise manner by adding additional processes, enabling the understanding of the individual effects of each process on the system [73]. Consequently, we construct a semi-idealized setup which allows the simulation results to clearly demonstrate only the effects of within-storm variability on the resulting hydrographs and flooded areas. Such a semi-idealized setup has previously been used to study the individual effects of processes on rainfall/runoff simulation results and is argued to ease the interpretation of the roles of different processes and catchment features in runoff generation [23,44,45]. In addition to this semi-idealized setup, we also compare results to simulations considering infiltration and heterogeneous roughness.

### 2.3.2. Boundary Condition Rainfall

In order to create an allochthonous flash flood scenario in the urban areas of the catchment (the central and lower areas), rain is imposed only to the upper subdomain which represents the mountainous sections of the area. This is consistent with previous reports on flooding in Tehran city as a result of precipitation in the upper regions of the catchment, and also with reports of the upper areas experiencing significantly higher rainfall compared to the lowlands [46,48,49].

Synthetic hourly rain distributions generated using a microcanonical random cascade model [74] are used for this study. The key basis of the cascade model is to gradually increase the temporal resolution of a certain rainfall depth while conserving mass and hyetograph general statistics in every step. The required disaggregation parameters are determined by stepwise aggregating observed hourly precipitation timesteps of identified storm events (throughout the input precipitation time series) to coarser temporal resolutions. From the precipitation time series, rainfall statistics can be derived and storms with particular features can be selected. In the next step, for a selected storm, a disaggregation procedure is carried out down to the desired temporal resolution (in our case, hourly). For more detailed information on the model and the parameterization procedure, the reader is directed to Pohle et al. [74].

Although a few meteorological gauge stations are available in the catchment area [75], the recordings of these gauges are not publicly accessible and could not be used for this study. Only IDF curves presented by Yazdi et al. [75] which are derived from the observation of those gauges were available. Due to this lack of data and difficulties in parameterizing the cascade model for the Kan catchment, a location in California, U.S.A. (Stockton, CA), with the same type of cold semi-arid climate (BSk) as Tehran city was selected as a plausible substitute. Hourly recorded rainfall recordings form the mentioned area (Stockton, CA) for a duration between 1975 to 2020 were used as input for the rainfall model parameterization, i.e., they were aggregated and parameters were estimated. However, this time series is not used to determine design storms (contrary to the original procedure to Pohle et al. [74]), since it is not a local time series. Instead, IDF curves from the region [75] are used to select a return period and the corresponding storm duration and depths. Using these rainfall features, we perform the disaggregation step.

Three different return periods (10, 100, and 1000 yearly) and two different durations (8 h and 16 h), resulting in 6 different events, were chosen. For each event, 49 different rain distributions are generated using the cascade model, which in addition to a single continuous block rain, result in 50 different rain signals per event, which have the same duration and volume, but only differ in the variability of the hyetograph. In total, 300 unique synthetic rains are used and imposed to the hydrodynamic model. The 8 h rains with return periods (RP) of 10, 100, and 1000 have depths of 52.8, 77.6, and 102.4 mm, respectively. For the 16 h rain signals, the depths are 67.2, 99.2, and 129.6 mm, respectively.

In order to be able to specify each rain signal and to indicate the temporal variability of the hyetographs, we use variance (typically represented as $\sigma^2$), which is computed as follows:

$$\sigma^2 = \frac{\sum(X - \mu)^2}{N} \tag{2}$$

in which $X$ represents each value in the dataset, $\mu$ represents the mean of all values, and $N$ is the number of data points. Each rain distribution is specified by the variance of its hyetograph. As examples, the hyetographs of three different rains of each event are illustrated in Figure 3. The blue line indicates the signals with $\sigma^2 = 0$, which is a simple continuous block rain, typically used as design rain for flood protection measures [15]. The red and green lines indicate two other rain signals of each event. In general, it is seen that rain signals with higher variances ($\sigma^2$) tend to have higher peak intensities. We emphasize that the generated rains of each event have the same duration and volume, making them completely comparable.

To get an overview on general properties of the rain signals, Figure 4 presents peak intensity (left column) and time of peak intensity (middle column) of all 300 rains. Additionally, to see how volume is distributed in each hyetograph, the volume of the first half of each rain signal divided by the second half is also presented (right column). There is a trend for rain hyetographs with higher $\sigma^2$, which tend to have higher peaks. Moreover, rains with larger return periods (e.g., Figure 4h) tend to have higher peak intensities compared to smaller return periods (e.g., Figure 4a). The time of peak intensity varies among different rains of the 6 events, with hyetographs both with peaks in the early stages and in the late

stages being present. Moreover, hyetographs with different distributions of volume, both the rain volume being concentrated in the first half or in the second half or the volume being relatively evenly distributed during the storm, are present in the generated rains.

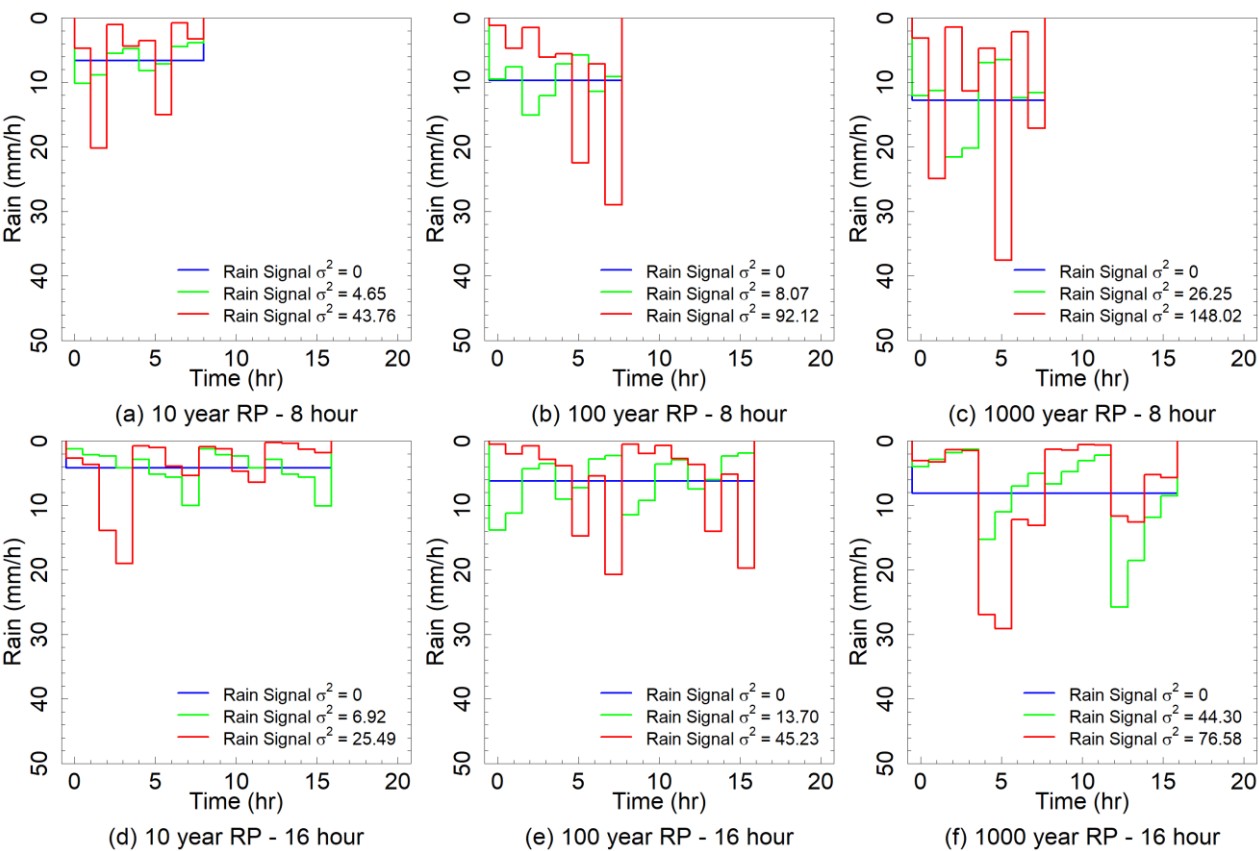

**Figure 3.** Examples of generated rain signals for each of the six chosen events ((**a**–**f**): 10, 100, 1000 RP—8 h, 16 h).

Within-storm variability might also consist of rainfall spatial variability. In order to explore how the spatial variability of rainfall may influence the results, we run some selected simulations with spatially variated rain fields. For this, rain signals of the 100-year RP 8 h event are used. Although some sophisticated methods for adding spatial variability to rainfall such as spatial interpolation [76] or orographic analysis [77] exist, they require multiple measurement points which, in our case, are not available. Further on, it is known that in many semi-arid regions such as Utah [78], Nevada [79], and Southern California [80], precipitation linearly increases with elevation. It has also been pointed out that a simple linear relationship between elevation and precipitation can be an acceptable approximation in most situations [81]. Consequently, we take the simplifying assumption that the relationship between rain intensity and elevation is linear. We generate rain fields of the 50 rain signals of the 100-year RP 8 h event under three different resolutions, dx = 0.5, 1, and 5 km. Rain intensity in each cell in each timestep is calculated as:

$$r_i = \frac{e_i}{S} \times n \times r_u \qquad (3)$$

in which $r_i$ (mm/h) is the rain intensity in cell $i$ in a specific timestep, $e_i$ (m) represents elevation of cell $i$, $S$ (m) is the sum of elevations of all cells in the rain induced domain, $n$ (-) is the number of cells, and $r_u$ (mm/h) is the rain intensity of the spatially uniform rain in that timestep. Elevation in the upper subdomain was upscaled to resolutions dx = 0.5, 1, and 5 km using arithmetic averaging of the original 12 m DEM for rain intensity

calculation in the spatially variated scenarios. Fifty rain signals of the 100-year RP 8 h event in combination with three resolutions resulted in 150 simulations under spatially variated rain fields. We emphasize that rainfall volume is conserved and is equal between all the spatially distributed scenarios and also the uniform setup. As a result, the rain setups are fully comparable in terms of rainfall volume. This avoids contradictory conclusions in assessing the effects of rainfall variability between different studies with difficulties to guarantee consistent rainfall volumes [18].

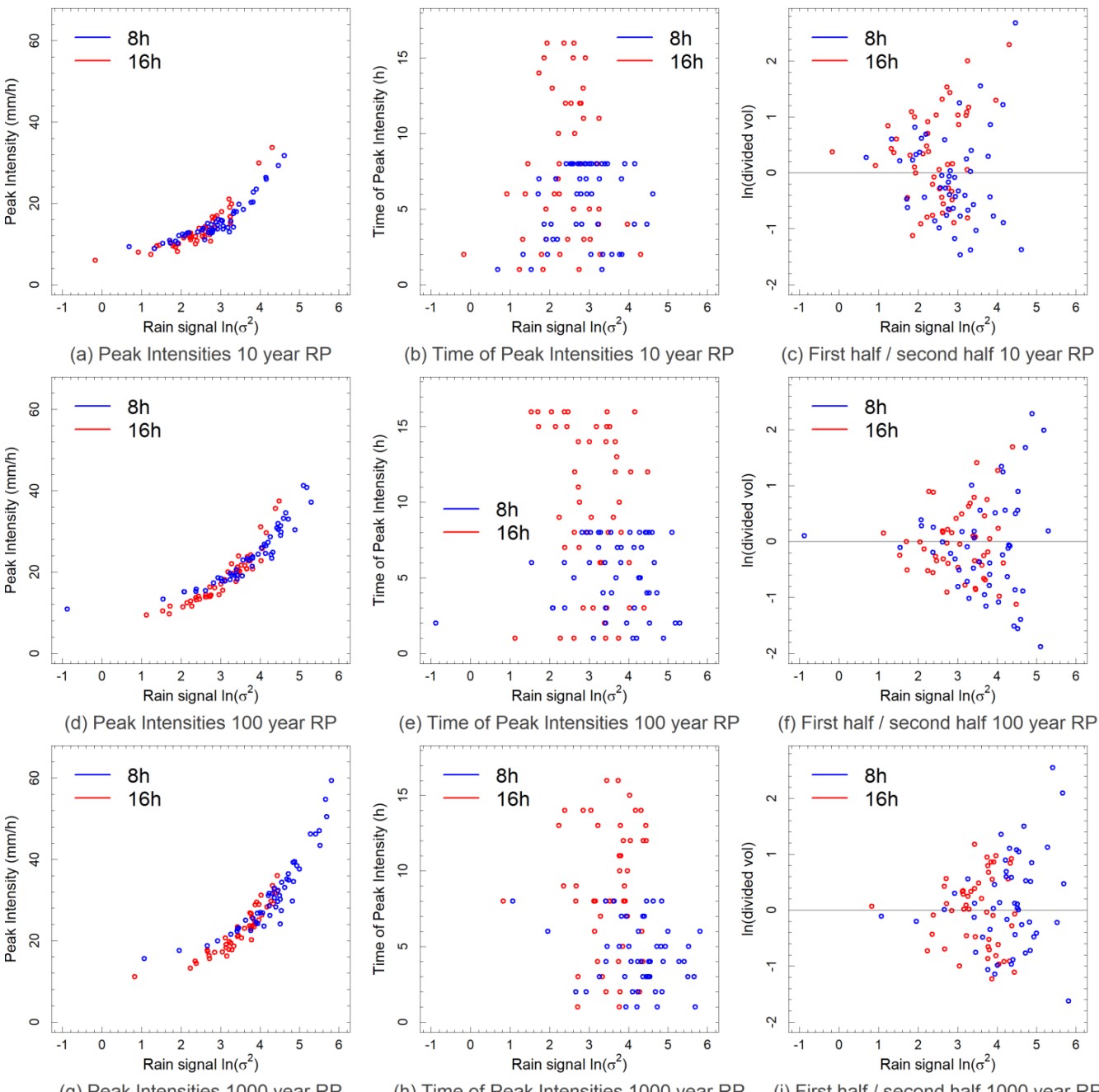

**Figure 4.** Peak intensity (**a**,**d**,**g**), time of peak intensity (**b**,**e**,**h**), and the volume of the first half of the hyetograph divided by the volume of the second half (**c**,**f**,**i**) for all 300 generated rain signals of the six chosen events (10, 100, 1000 RP—8 h, 16 h). The x axis of the figures and the y axis of the figures in the third column are plotted with a natural logarithmic scale.

## 3. Results and Discussion

First, we begin with the analysis of the effects of temporal within-storm variability. Some examples of the resulting hydrographs at measuring point A (Figure 2) for the spatially uniform rain scenarios are presented in Figure 5 in a similar order as the hyetographs represented in Figure 3. In general, the main shape of the hydrographs is governed by the shapes of the rain hyetographs. In most cases, rain signals with higher variances produce higher peak discharges, which is due to their higher peak intensities. Interestingly, in some cases, such as the 1000-year RP 16 h event, rain with higher peak intensities do not necessarily result in higher peak discharge. As an example, in Figure 3f, whereas rain $\sigma^2 = 76.58$ (red line) has a higher peak intensity and also a higher hyetograph variability in comparison to rain $\sigma^2 = 44.30$ (green line), rain $\sigma^2 = 44.30$ tends to produce a higher peak discharge by approx. 20% (Figure 5f). The same behavior can also be seen among rains $\sigma^2 = 4.65$ and 43.76 (Figure 3a) for the 10-year RP 8 h event rains.

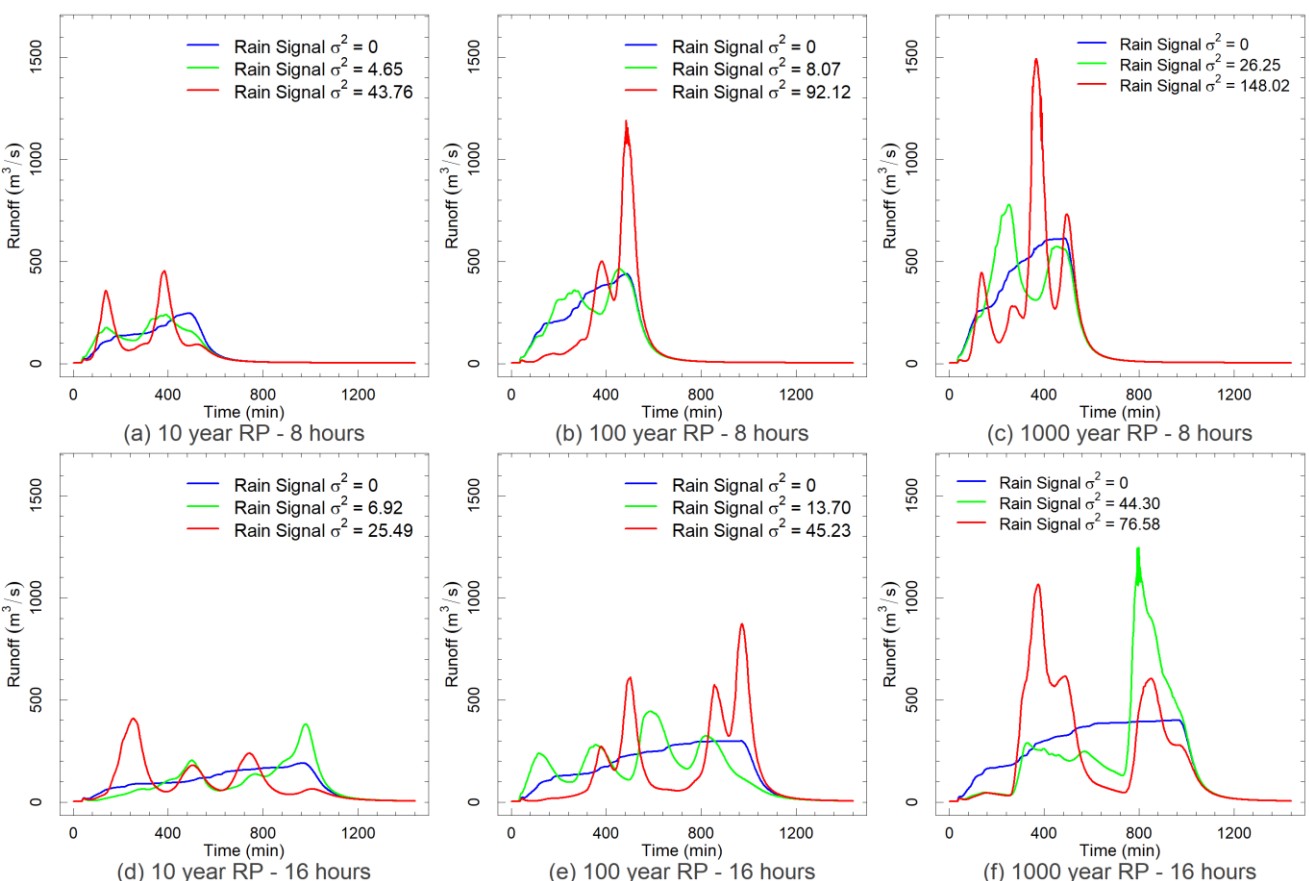

**Figure 5.** Example hydrographs at measuring point A for rains presented in Figure 3 under a spatial uniform setup. ((**a**–**f**): 10, 100, 1000 RP—8 h, 16 h).

To see how peak discharge is affected by within-storm temporal variability, Figure 6 presents the measured peak discharges at measuring points A, B, and C (Figure 2) for all rain signals of the six selected events. Peak discharge of the $\sigma^2 = 0$ rain of each event is plotted as a line to act as reference. In most cases, the highest peak discharge is achieved at measuring point A, which is located in the main Kan River, just downstream of the rain-induced area (upper subdomain, Figure 2). At this measuring point, higher variated rain hyetographs tend to produce higher peak discharges, which is mainly the result of their higher peak rain intensities. However, this is not the case for measuring points B and C. These points measure the flow after passing both rain-induced and non-rain-induced areas in the catchment. Different rain signals, despite having different peak intensities, roughly produce peak discharges in the same range. This clearly indicates the retention

effect of the river channel and its adjacent flood plains in the catchment, which highly filter out the effect of hyetograph variability and peak intensity of the rain signal on discharge. This also highlights the importance of the measurement location of discharge in the river channel. Implementing design measures in the lower parts of the catchment based on measured discharge at the upper catchment (point A) could result in an overdesign and cost inefficiency. Another indication from the results is that in river channels with low retention capacities (e.g., steep artificial rectangular channels), higher peak discharge values are achieved by higher temporal variated rains. However, in channels with high retention capacities, such as natural rivers, the effect of rainfall temporal variability on the resulting discharge would be highly filtered out.

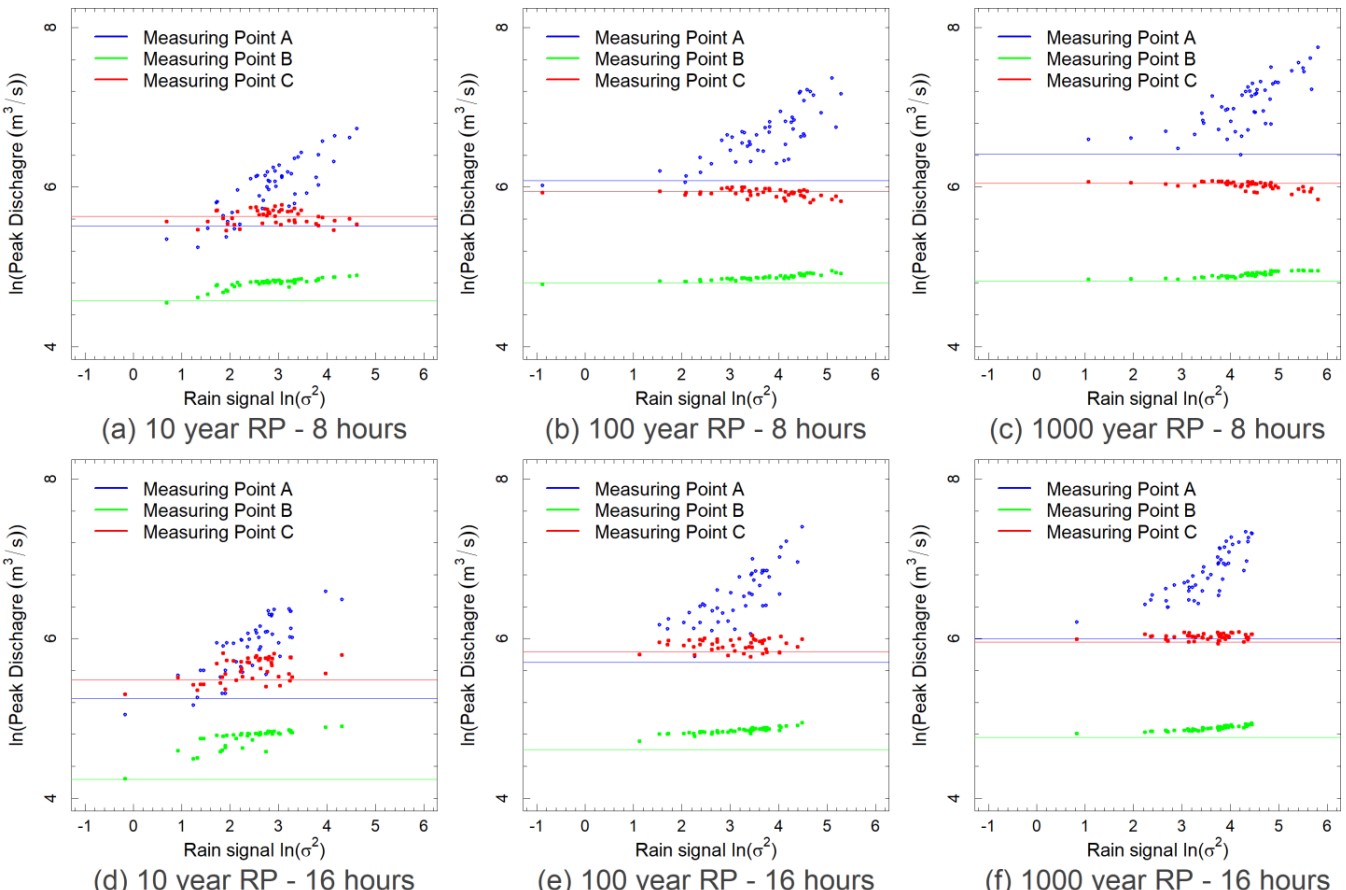

**Figure 6.** Peak discharges at measuring points A, B, and C for rain signals of the 10-, 100-, and 1000-year RP events with durations of 8 or 16 h. Both the x and y axes are plotted using a natural logarithmic scale. The peak discharge of rain variation $\sigma^2 = 0$ of each event is plotted as a line. ((**a**–**f**): 10, 100, 1000 RP—8 h, 16 h).

The general behaviors among the six events seem to be relatively the same. Higher return period events are seen to produce higher peaks, mainly due to their higher rain volumes and intensities. However, the effects of rainfall temporal variability show no sensitivity to different return periods, with differences in peak discharge among different rain signals being relatively the same among the three return periods. The resulting peak discharge values in point A, among all six events, indicate that rainfall temporal variability can highly influence peak discharge in rain-induced areas. This is in line with previous findings such as Faures et al., [25] and Paschalis et al. [3] which also highlighted the great influence of rain temporal variability on generated peak discharge.

Discharge alone can be an insufficient indicator for the severity of a flood. Consequently, we study the behavior of flooded areas with water levels above 10 and 50 cm in

Figure 7. In general, it can be seen that higher return period storms achieve larger flooded areas in comparison to storms with lower return periods, attributable to the higher intensities and volumes of these storms. Storm duration also shows to not be much of an influence in the resulting flooded areas. On the other hand, the variability of the hyetographs is seen to be a governing factor. An overall increase is present in all six events in the flooded areas by an increase in rainfall variability. This highlights the importance of rainfall temporal structure on the resulting flooded areas. As an example, a high variated hyetograph of a 100-year RP event can result in larger flooded areas in comparison to a low variated hyetograph of an 1000 year RP event, which highlights the importance of adequately representing rainfall temporal structure and variability in flood mitigation design.

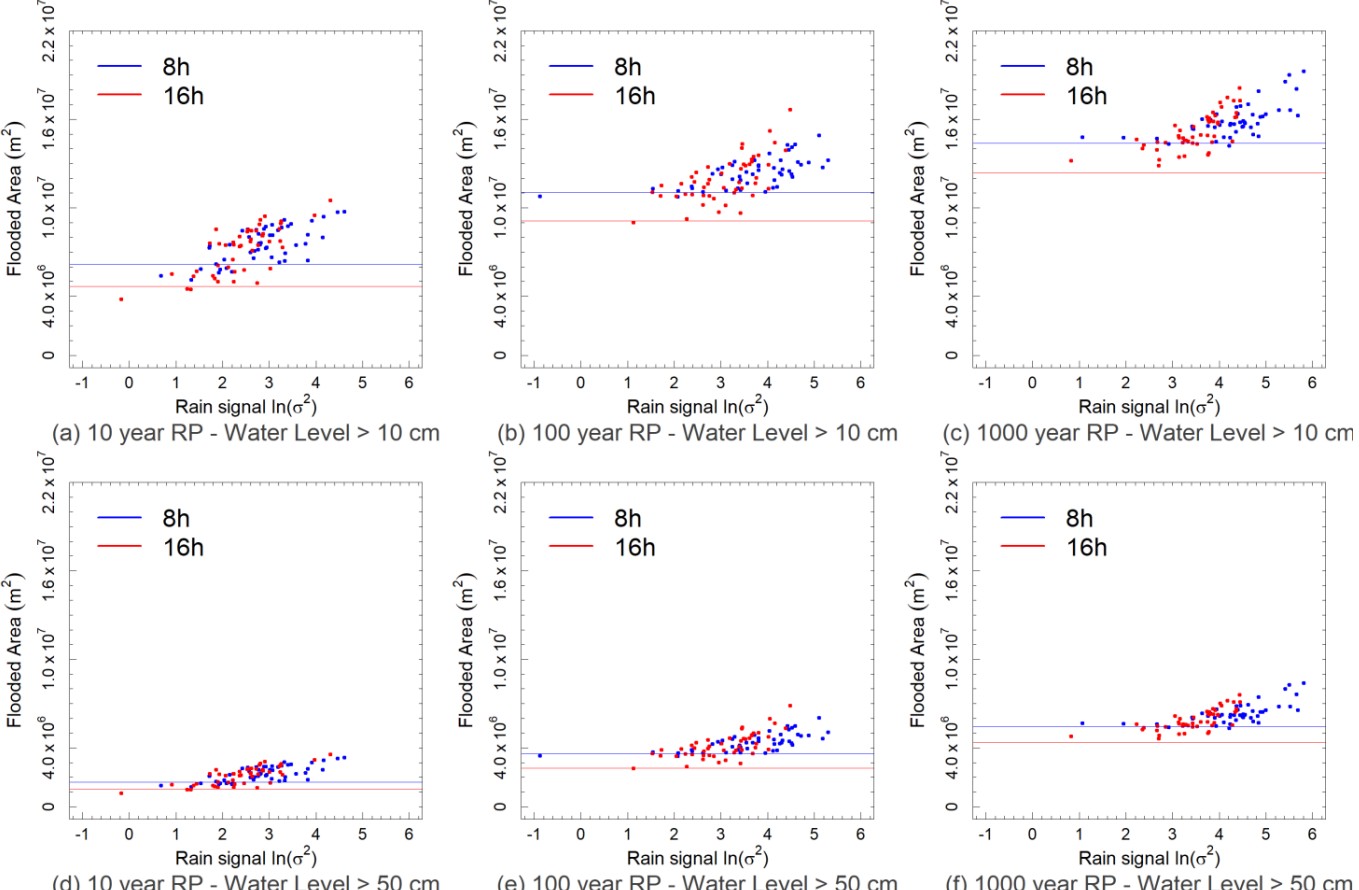

**Figure 7.** Flooded areas with water levels above 10 and 50 cm for the rain signals of the six selected events. The x axes of the figures are plotted with a natural logarithmic scale. The flooded areas of rain $\sigma^2 = 0$ of each event are plotted as a line. ((**a**–**f**): 10, 100, 1000 Water Level—10 cm, 50 cm).

The flooded areas with water levels above 10 cm for the 100-year RP 8 h rains are shown in Figure 8. The flooded areas are plotted against rainfall peak intensity (Figure 8a) and measured peak discharges in measuring point A, B, and C (Figure 8b–d). Additionally, a linear regression analysis has been performed on each plot to indicate how correlated the variables are with one another. The flooded area is seen to be mostly correlated to peak discharge measured at measuring point A, which is located in the main Kan River at the lower part of the upper subdomain. However, it is seen that peak discharge at measuring point C shows the least correlation with flooded areas. This indicates again how the river channels in areas not receiving rain in the catchment may act as a buffer and reduce the relevance of discharge on flooded areas by their retention capacity. It also highlights the importance of discharge measuring location in the catchment for flood mitigation design. Additionally, rainfall peak intensity is also shown not to be strongly

correlated with flooded areas, indicating the importance of other factors such as hyetograph shape and also catchment topography in surface runoff generation.

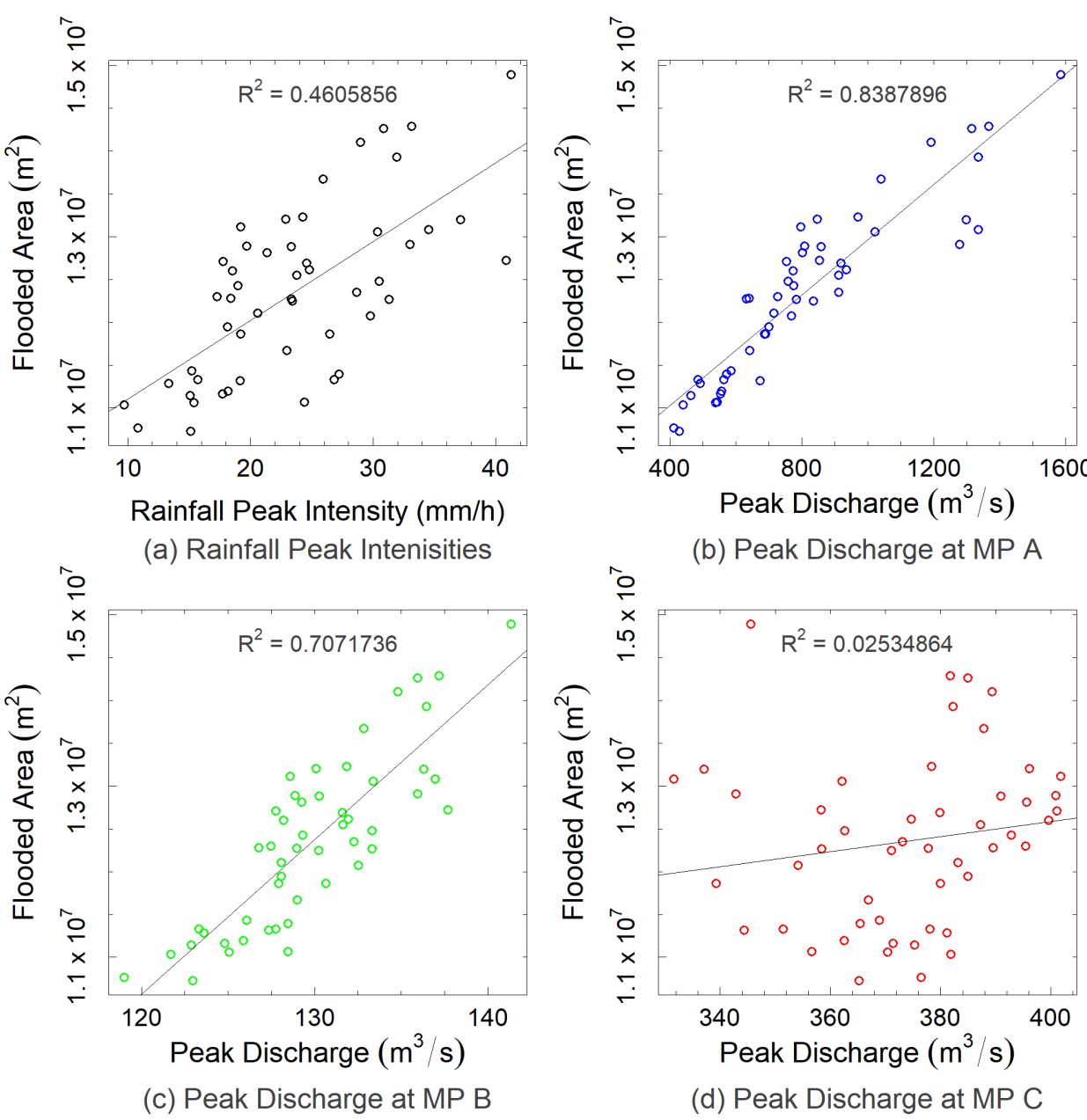

**Figure 8.** Flooded areas in the two lower subdomains with water levels above 10 cm plotted against rainfall peak intensity and measured peak discharges at measuring points A, B, and C for the rain signals of the 8 h 100-year RP event. (**a**) Rainfall Peak Intenisities. (**b**) Peak Discharge at MPA. (**c**) Peak Discharge at MPB. (**d**) Peak Discharge at MPC.

A second part of the analysis investigates the effects of spatial variability of precipitation. Three examples of computed hydrographs at measuring point A for the rains $\sigma^2 = 0$, 8.07, and 92.12 (Figure 3b, 8 h 100 RP event) under spatial variations of dx = 0.5, 1, and 5 km and also a uniform spatial distribution are presented in Figure 9. The hydrographs seem to be very similar with very mild differences seen in the onsets and also the peaks of the hydrographs. The uniform rainfall represents the scenario with no spatial variability. Spatially variated scenarios with higher spatial resolutions (for example, dx = 0.5 km) represent scenarios with higher spatial variability due to their higher number of cells.

The spatial variance ($\sigma^2_s$) of rainfall intensities during the 8 h rainfall in the simulations for the spatially distributed and also uniform scenarios is presented in Figure 10. The presented spatial variance acts as indicator of the degree of rainfall spatial variability in the different scenarios with higher values indicating higher spatial variability of the rainfall. Clear differences are seen in spatial variability between the scenarios. Higher resolution distributions show higher variability in comparison to lower resolution rains and also the uniform scenario. Higher temporal variated rains ($\sigma^2 = 92.12$) also achieve higher spatial variance in comparison to lower temporal variated rain ($\sigma^2 = 0$). The highest variances are also achieved the times in which higher intensities are present.

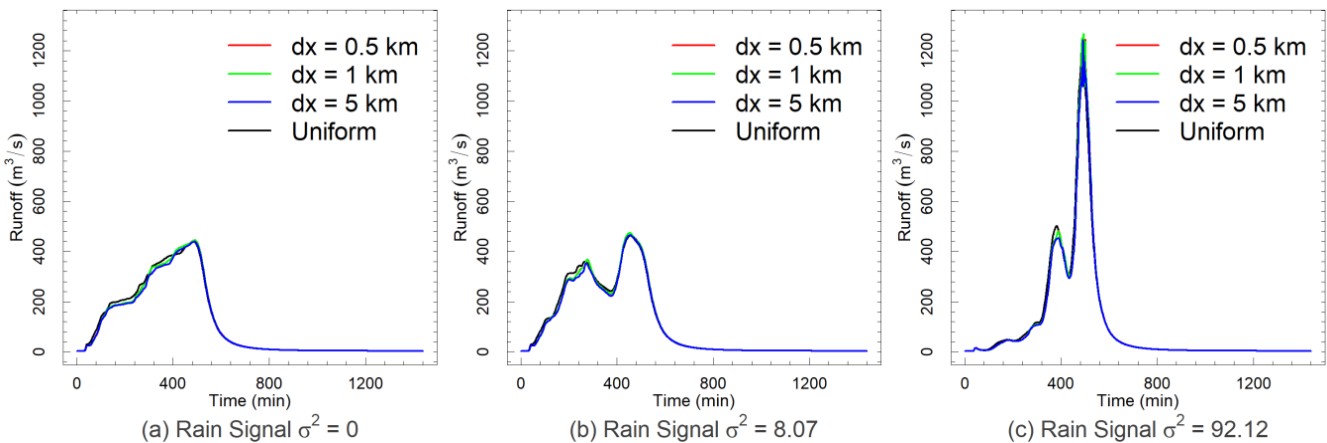

**Figure 9.** Example hydrographs at measuring point A for rains $\sigma^2 = 0$, 8.07, and 92.12 (**a**–**c**) of the 100-year return period 8 h event under the uniform spatial distribution and also spatially variations of dx = 0.5, 1, and 5 km.

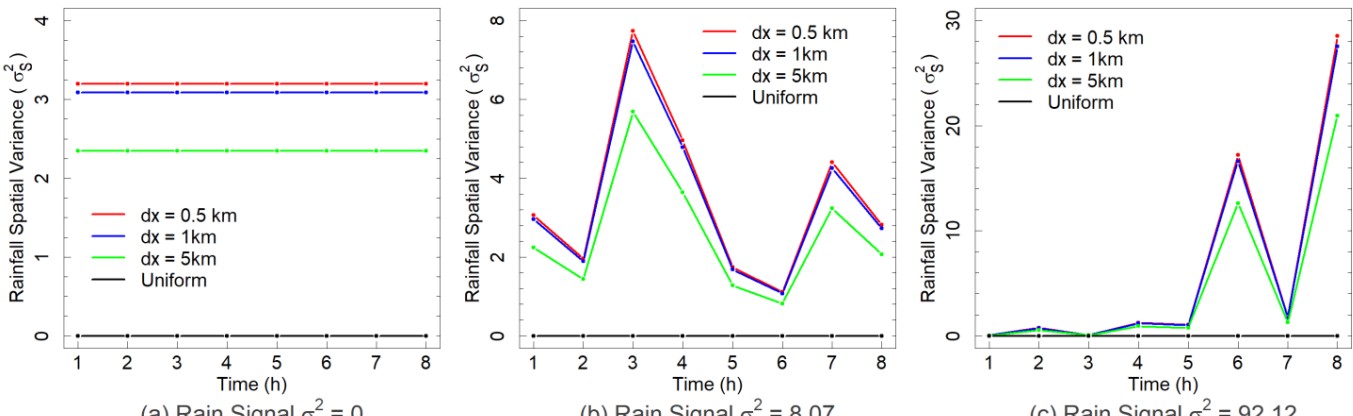

**Figure 10.** Spatial variance of rain intensity for rains $\sigma^2 = 0$, 8.07, and 92.12 (**a**–**c**) of the 100-year return period 8 h event under the uniform spatial distribution and also spatial variations of dx = 0.5, 1, and 5 km.

To get a better comparative view on how spatial variability affects the hydrographs, Figure 11 presents peak discharges and time of peak discharge for the three spatially variated resolutions normalized by the uniform setup at all three measurement points. In general, the effects of spatial variability on runoff are seen to be lower in comparison to temporal variability. Spatially variated rain results in differences in peak discharge up to 10% in measuring point A. However, at the other two locations, the differences are very low in around 2–3%. The time of peak discharge is also seen to be mildly affected by spatial variability, with maximum differences of 10 percent in the three measurement points. These findings are highly in line with works such as Smith et al. [30] and Paschalis et al. [3] which

observed, in their case studies, that peak discharge and peak timing are less sensitive to rainfall spatial variability in comparison to rainfall temporal variability.

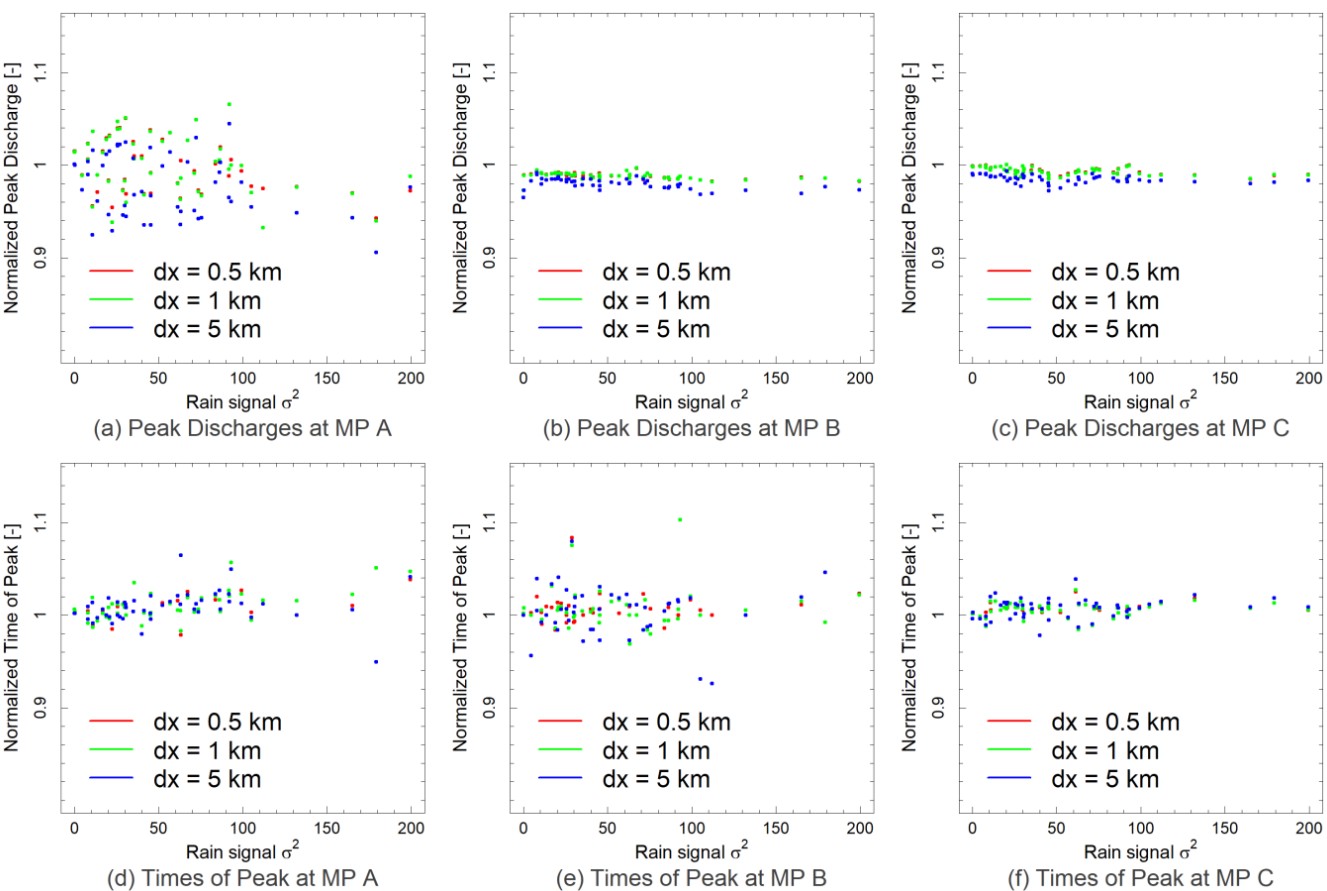

**Figure 11.** Peak discharge and time of peak discharge normalized by the uniform distribution setup for 50 8 h rains with a 100-year RP with spatial distributions of dx = 0.5, 1, and 5 km at measuring point A, B, and C. (**a**) Peak Discharge at MPA. (**b**) Peak Discharge at MPB. (**c**) Peak Discharge at MPC (**d**) Times of Peak at MPA. (**e**) Times of Peak at MPB. (**f**) Times of Peak at MPC.

Flooded areas with water levels above 10 and 100 cm for the spatially distributed setups (dx = 0.5, 1, and 5 km) normalized by the uniform scenario are presented in Figure 12. The observed differences between the uniform and the spatially variated scenarios are less than 10%. In most cases, spatially variated storms produce less floodings compared to uniform rain. It is also seen that whereas higher variated (resolution) cases such as dx = 0.5 and 1 km show differences of up to 2% with the uniform rain setup, the less spatially variated rain (dx = 5 km) shows higher differences of up to 5–6%. One reason for this might be that higher variated rain (higher resolution) tends to have higher maxima intensities in the local maxima of the topography, whereas in the lower variated (resolution) setups, due to the averaging effect of coarsening, lower intensities are present. However, it should be pointed out that the low variated (resolution) rain still has a variated rain field and differs from the uniform setup. The temporal variability of the rain also does not interact with spatial variability, with the differences between different resolutions of the different rain signals being relatively in the same range.

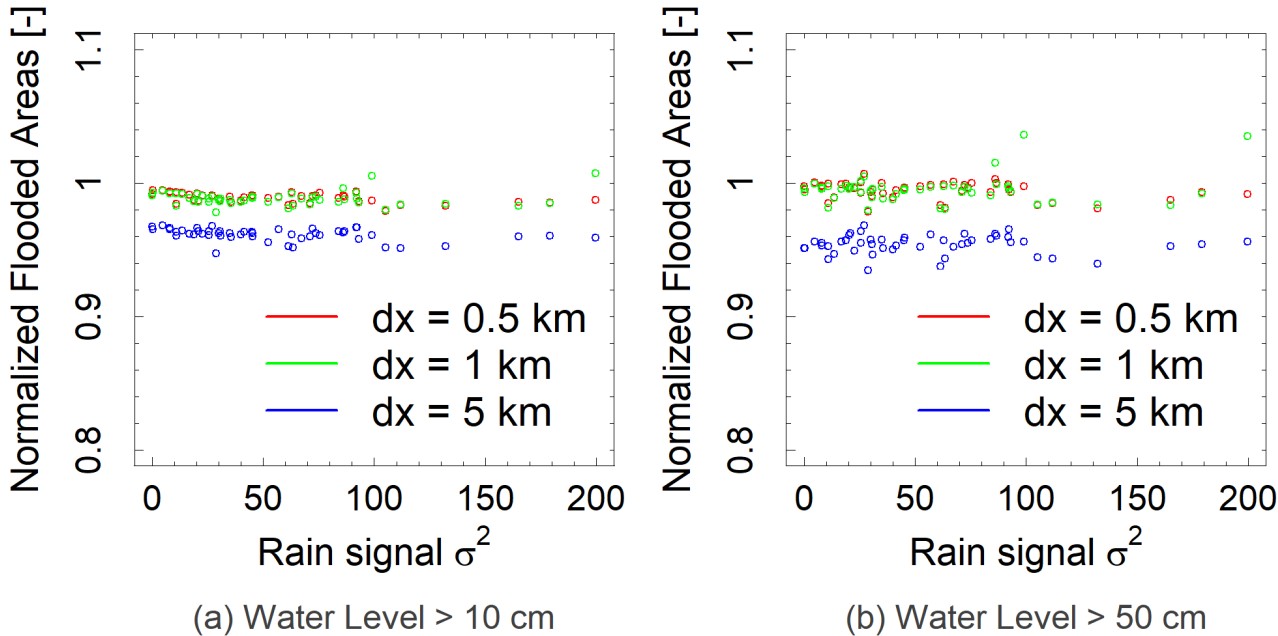

**Figure 12.** Flooded areas normalized by the uniform distribution setup with a water depth exceeding 10 cm (**a**) and 50 cm (**b**) under spatial distributions dx = 0.5 km, dx = 1 km, dx = 5 km for the 50 rain variations of the 8 h 100-year RP storm.

In order to assess the robustness of our results to the initial simplifying assumptions on roughness and infiltration, an additional number of simulations under a non-idealized setup with heterogeneous roughness and a uniform infiltration were run. The fifty rain signals of the 100-year RP 8 h event under a uniform spatial rain are used for this comparison. Figure 13a,b present a comparison between hydrographs for rains $\sigma^2 = 0$, 8.07, and 92.12 (Figure 3b) under the idealized and non-idealized setups, respectively. Additionally, a comparison of peak discharge and flooded areas with water levels above 10 cm for the 50 rain signals under the idealized and non-idealized setups are shown in Figure 13c,d.

Roughness in the non-idealized setup is based on the land use map presented by Ghorbanian et al. [82] from the catchment area. Manning values presented by Papaioannou et al. [83] are applied for each land use category, respectively. Infiltration is calculated by the Soil Conservation Service's (SCS) Curve Number (CN) approach [84]. Cumulative rainfall for a certain rain signal at each time interval is determined by the following:

$$Q = \frac{(P - I_a)^2}{P - I_a + S} \tag{4}$$

where $Q$ is cumulative excess rainfall, $P$ represents cumulative rainfall depth, $I_a = \min(P, 0.2S)$ is the initial abstraction, and $S$ represents the potential maximum retention which, if millimeters are used as units for rainfall depth, is calculated as:

$$S = 25.4 \left( \frac{1000}{CN} - 10 \right) \tag{5}$$

$CN$ was homogenously set to 81 for the upper catchment area. This $CN$ value is suitable for arid and semiarid rangelands [85]. Although the assumption of a homogeneous infiltration could be a strong simplification, the main objective of running these simulations is comparing to an impervious case, and the next simplest representation of infiltration (uniform) suffices. Though the implementation of more sophisticated infiltration models such as Horton and Green-Ampt is possible, this would open the parameterization space [86], which is not the main focus of this study. The significance of implementing a stepwise ap-

proach in including additional processes has been previously pointed out [44,62,65,87,88], as it facilitates understanding how processes manifest in the hydrological signatures.

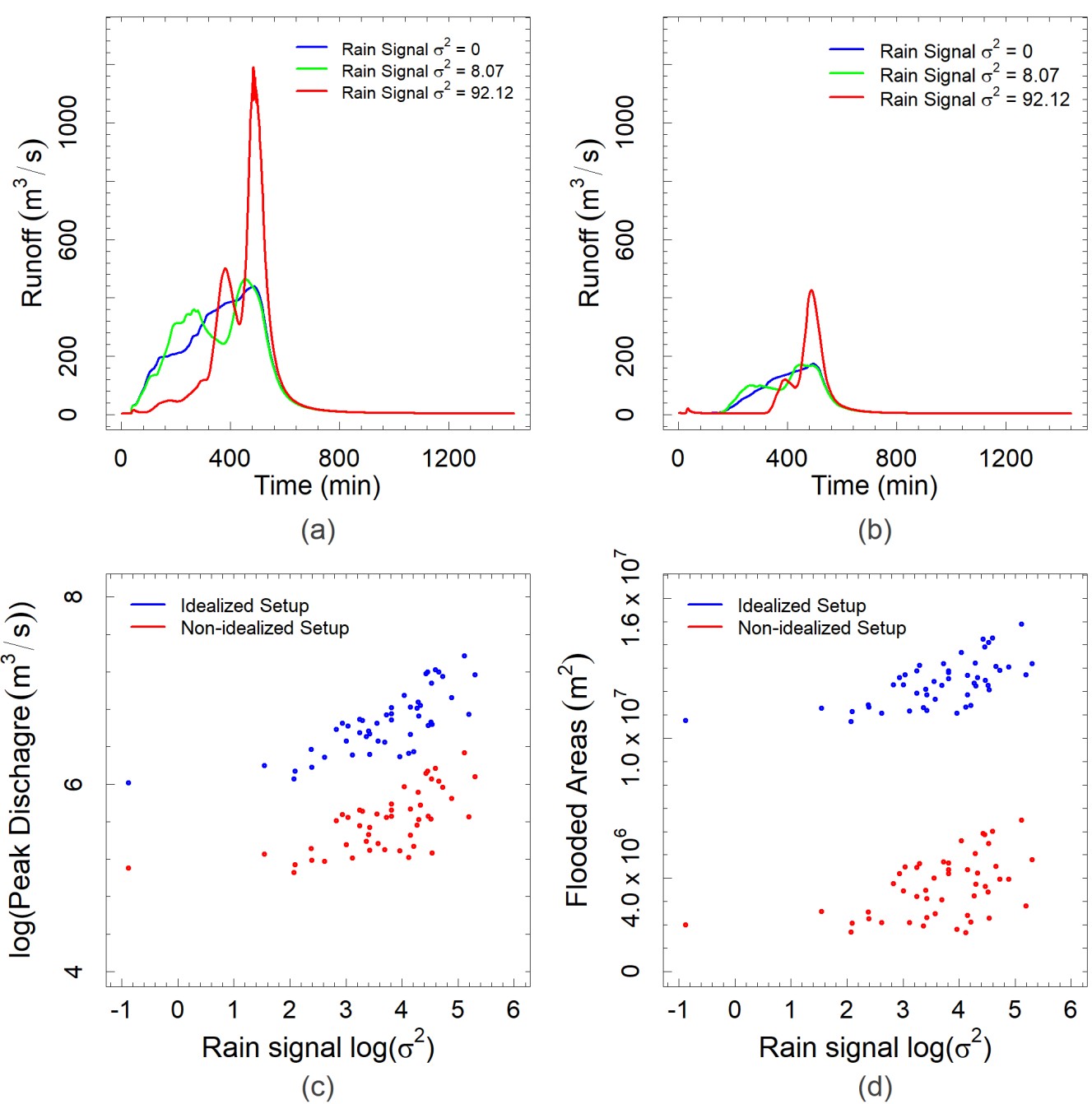

**Figure 13.** Example hydrographs at measuring point A for rains $\sigma^2 = 0$, 8.07, and 92.12 (Figure 3b) of the 100-year RP 8 h event under the idealized (**a**) and non-idealized (**b**) setups. Peak discharges at measuring point A (**c**) and flooded areas (**d**) for all 50 rains under the idealized and non-idealized setups are presented in c and d, respectively.

A dampening effect is seen in the hydrographs as a result of roughness and infiltration in Figure 13a,b. This effect is seen clearly as lower peak discharges and delayed onsets of runoff. However, the differences among the three hydrographs remains relatively the same, but are scaled down due to the change in the magnitude of runoff. This behavior can clearly be observed in Figure 13c, which presents measured peak discharges at measuring point A for the 50 rain signals under the idealized and non-idolized setups. Whereas the

magnitude of peak discharge is reduced in the non-idealized setup, the differences among the rains remain mostly unchanged. The differences in flooded areas also follow the same trend, with flooded areas only being reduced in magnitude in the non-idealized setup in comparison to the idealized setup. However, a very mild spread is seen in the flooded areas of the non-idealized setup compared to the idealized setup. This may hint that flooded areas under different rainfall temporal variabilities are more sensitive to infiltration and roughness in comparison to discharge.

## 4. Conclusions

The objective of this work was to study how within-storm variability can affect rainfall/runoff generation and the resulting flooding. Hydrodynamic modelling of the rainfall-runoff process was carried out for the Kan catchment (Iran), forced by synthetic rain hyetographs with different degrees of spatiotemporal variabilities. The effects of within-storm variability were assessed in terms of runoff generation and flash flood extent. Six events with different return periods and durations were chosen and, for each event, a set of fifty hyetographs with different temporal variabilities were generated using a microcanonical random cascade model. Additionally, a set of simulations with different levels of spatial variability were run and compared.

The results show that within-storm variability has a strong effect on the runoff generation process and on the resulting flooded areas. In particular, both the peak discharge and the early stages of runoff—the onset of runoff—are affected. Flooded areas are also sensitive to both the spatial and temporal variability of rainfall. Storms with higher temporal variability generally resulted in higher peaks and larger flooded areas. The spatial variability of rainfall reduced the flooded areas in comparison to spatially uniform rainfall. The results for the Kan catchment suggest that discharge and flooded area are more sensitive to temporal within-storm variability than to spatial variability of rainfall. Temporal within-storm variability was also shown to be a larger influence on runoff and flooding in comparison to storm properties such as return period, volume, and duration, highlighting the importance of taking it into account and an adequate rainfall structure representation in flood design/planning management. Land use roughness variability, at least in the case of the Kan catchment, was seen to only have very mild influence on the effects of storm temporal variability in the simulations.

Furthermore, the sensitivity of discharge in the stream, in response to different rainfall temporal and spatial distributions, depends on the location along the stream. For allochthonous streams, this means the distance to the region experiencing rainfall. Upstream reaches appear to be more sensitive than lower reaches due to both in-channel and floodplain retention effects, and also likely influenced by the asynchronous superposition of hydrographs resulting in small mountainous valleys. This may provide some practical guidelines to assess the priority of how much attention needs to be given to within-storm variability in case studies; that is, in the analysis of catchments with high retention capacity, the relevance of within-storm variability is likely to be at a low level. This also indicates the importance of preserving river channels retention capacity and taking it into account as part of flood mitigation measures.

Though this study offers insight into the effects of rainfall spatiotemporal variability by means of hydrodynamic modelling, given that every basin is unique [89], further studies on other catchments with different conditions would be beneficial for generalization of the results. Furthermore, the evaluation of rainfall spatial representations techniques such as traditional spatial interpolation, remote sensing retrieval, machine learning methods (RF, ANN, and SVM), and multi-source rainfall merging would be worthwhile, which are shown to have important influence on rainfall representation [90,91]. The simplifying assumptions in this work, as well as the limitations in the available catchment, also remind of the need to explore how rainfall variability interacts with heterogeneous morphological and land use catchment properties, such as vegetation cover [92,93], climate [94], topography [45], and microtopography [95] on the runoff generation process. This may be particularly

relevant in semi-arid regions where rainfall variability interacts with spatial heterogeneity (e.g., microtopography and vegetation patches) to produce complex connectivity and flow paths [96] and runoff scale effects as complex as a reduction in runoff downslope [39,97], which, in our setup, have not been explored. A better representation of river bathymetry in the simulations, which is known to significantly affect channel storage [72], is further recommended for future works, which should be pointed out as a limitation of this study.

Furthermore, the results of this study highlight the importance of taking into account different rainfall characteristics into flood modelling. This warrants further studies on other rainfall properties such as storm dynamics (movement, direction, speed etc.), which have been hinted to be a crucial influence on a catchments flood response and, in particular, on hydrograph timing [23,98].

**Author Contributions:** Conceptualization, S.K.B.G. and D.C.-V.; methodology, S.K.B.G.; software, D.B. and S.K.B.G.; validation, S.K.B.G.; formal analysis, S.K.B.G. and D.C.-V.; investigation, S.K.B.G.; writing—original draft preparation, S.K.B.G.; writing—review and editing, S.K.B.G., D.B., D.C.-V. and C.H.; visualization, S.K.B.G.; supervision, D.B., D.C.-V. and C.H.; project administration, D.B.; funding acquisition, D.B. All authors have read and agreed to the published version of the manuscript.

**Funding:** This research was funded by Federal Ministry of Education and Research (BMBF) with the grant number 13N15180 under the project HOWAMAN.

**Data Availability Statement:** Not applicable.

**Acknowledgments:** This research was part of the project HoWaMan (FKZ: 13N15180), which was supported by the German Federal Ministry of Education and Research (BMBF) in the "International disaster and risk management" call (IKARIM) as part of the "Research for Civil Security" framework program.

**Conflicts of Interest:** The authors declare no conflict of interest.

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
