# Peer review of "Effects of Within-Storm Variability on Allochthonous Flash Flooding: A Synthetic Study"

_water, doi:10.3390/w15040645_

Round 1
Author Response
Dear reviewer,
Thank you for reading our manuscript and for your feedback and comments. Please find below our replies to your comments:
Regarding comment 1, Thank you for the comment. A sentence explaining the reason for the choice has been added in line 219.
Regarding comment 2, An brief explanation in addition to a new citation has been added in line 304.
Regarding comment 3, Thank you for the comments. Some sentences have been added to make it more clearer for the readers in lines 328 and 342. The return period precipitation durations and depths are obtained from IDF curves from the region. These storm properties are then disaggregated into a higher resolution temporal series.
Regarding comment 4, Thanks for pointing this out. The change has been implemented.
Regarding comment 5, Thanks for the notice. The notations in all figures throughout the paper have been corrected.
Regarding comment 6, Some words have been added to the sentence to make it more clear in line 513. So in the second part that we investigate the effect of rainfall spatial variability, we have one uniform scenario, which the rain has no spatial variability with only one cell covering all the upper domain, and we have three spatially variated scenarios with spatial resolutions of dx=0.5, 1 and 5km.
Regarding comment 7, Thanks for pointing this out. The methodology for variance calculation is the same. An equation for variance has been added in line 358. We have also renamed spatial variance to σ2s in the text and also figure 10, to be distinguishable from temporal variance for the reader.
Regarding comment 8, Sentence is rephrased (line 547).
Regarding comment 9, Thank you for the comment. More explanations on this have been added in line 550. Some unclarity has also been explained in comment 6. The methodology for flooded areas calculation has also been added in line 291
Regarding comment 10, Thank you for pointing it out. It is now corrected.
Reviewer 2 Report
The paper is written lucidly. It presents the results of a high quality scientific study and I recommend it for publication.
Author Response
Dear reviewer,
Thank you for taking the time to read our manuscript and for your feedback.
Reviewer 3 Report
Dear Authors,
The paper presents a topic of interest for researchers and practitioners. However, several improvements are needed.
1. Presentation of the need for this study.
2. The introduction should be supplemented with other relevant research conducted in this field. In this way the research will show what is the gap it fills and what are the elements of originality. The introduction section must present the structure of the paper and what this paper proposes. The introduction could be systematized to make it easier to follow.
5. The discussion section should be consolidated and present the main results. Comparisons should be made with other studies (numerous in this field).
6. To emphasize the need for this study.
7. The stages of the methodology must be presented in detail.
8. To highlight the gaps filled by the present study.
9. The conclusions section should be completed with a review of the research.
Author Response
Dear reviewer,
Thank you for reading our manuscript and for your feedback and comments. Please find below our replies to your comments:
Regarding comment 1, Thank you for the comment. Some sentences emphasizing the contributions of the study have been added in line 157.
“The main components of this study which build on previous works on the topic of rainfall spatiotemporal variability are as follows: 1. Studying the effects of rainfall spatiotemporal variability on flash flooding and runoff in an allochthonous environment, which has not received much attention until now. 2. construction of a controlled systematic modelling environment which enables studying the effects of rainfall spatiotemporal variability, in-dependent of other processes, on surface flow and flooding. This setup also allows for rainfall volume consistency among the simulations which enables straightforward comparability. 3. In addition to the analysis of the resulting hydrographs in the simulations, also taking into account other indicators such as flooded areas, which allow for a more comprehensive view and understanding on the effects of rainfall spatiotemporal variability.”
The need for this study has also been pointed out in several parts of the introduction such as lines 79-84, 115-117 122-129.
Regarding comment 2, Thank you for the suggestion. Some sentences explaining the structure of the paper have been added in line 176. An extensive literature review on the topic of rainfall spatiotemporal variability is also presented in the 5th paragraph of the introduction (line 85), citing and explaining several works done on this topic and also why further studies are necessary.
Regarding comment 5, Thank you for the comment. Some sentences comparing the results of this work with previous studies have been added to the paper, such as lines 448 and 527. Additionally, sentences explaining the results in more detailed have been added such as in line 539.
Regarding comment 6, Thank you for reemphasizing this point, which is similar to comment 1. This comment has been addressed in comment 1.
Regarding comment 7, Thank you for the comment. An explanation of the steps taken in the study are added in line 247. Some sentences explaining the methodology in more detail have also been added throughout the paper such as in lines 291, 304, 328 and 342.
Regarding comment 8, Thank you for reemphasizing this point which was also mentioned in comment 2. This point has mainly been addressed in comment 2.
Regarding comment 9, Thank you for the suggestion. A review of the work is presented in the first paragraph of the conclusion section in line 613.
Further on, the manuscript text has been thoroughly reviewed and English language corrections have been made.
Round 2
Reviewer 1 Report
The revision and reply are reasonable.
Reviewer 3 Report
I accept this version of the paper.